# Revisiting the Landscape Mosaic model

**Peter Vogt** [1]*, **James Wickham**[2], **José Ignacio Barredo** [1], **Kurt Riitters** [3]

**1** European Commission, Joint Research Centre, Ispra, Italy, **2** United States Environmental Protection Agency, Research Triangle Park, North Carolina, United States of America, **3** United States Department of Agriculture, Forest Service, Research Triangle Park, North Carolina, United States of America

* peter.vogt@ec.europa.eu

**Data Availability Statement:** All relevant data are within the manuscript and its Supporting Information files.

**Funding:** The author(s) received no specific funding for this work.

## Abstract

The landscape mosaic model quantifies and maps the spatial juxtaposition of different land uses. It provides a landscape perspective of anthropic threats posed by agriculture and urban development, and the spatial-temporal shifting of the landscape mosaic indicates landscapes where anthropic intensity has changed. We use the U.S. Geological Survey provided National Land Cover Database (NLCD) for the years 2001 and 2021 to derive the landscape mosaic at five analysis scales. To improve earlier implementations of the model, we introduce the heatmap, a flexible scheme providing more thematic reporting opportunities and allowing for better quantitative summary reporting across analysis scales as well as for temporal trends. The results are exemplified at regional scale for the Atlanta metropolitan area. We use the improved model to investigate the land cover context over time and at different analysis scales and show how custom color tables detail different thematic features of the landscape mosaic, including the degree and change of anthropic intensity. We conclude with a discussion of potential applications in ecology, landscape planning, and restoration to illustrate the benefits of the revised landscape mosaic model. All assessment tools are now available in open-source software packages.

## Introduction

Patterns of land cover are the result of naturally occurring phenomena, such as the intermingling of shrublands and forests, as well as human created patterns such as the spread of agriculture and urban development into previously undeveloped areas. Urban development and agriculture are activities unique to humans. Both activities not only destroy natural capital, which constrains provision of a range of ecosystem services, but also pose potential risks to a variety of ecosystem services in surrounding areas through mechanisms related to edge effects [1, 2] and matrix effects [3, 4]. Thus, measuring land cover or land use (https://www.sciencedirect.com/topics/earth-and-planetary-sciences/land-use-classification) has been a focal point of interest for decades to quantify the human footprint on our environment [5] and to monitor status and trends for environmental policy and landscape planning.

Many national and international assessments have used land cover maps derived from remotely sensed images which portray intuitive and relevant land cover classes (e.g., forest, agriculture, urban, water, grassland). The National Land Cover Database (NLCD) is a prime

**Competing interests:** The authors have declared that no competing interests exist.

example of consistent monitoring providing fully comparable datasets of land cover over time. The most recent NLCD land cover product [6] covers the conterminous United States from 2001 to 2021 at a spatial resolution of 30m x 30m (0.09 ha per pixel) for a 16-class legend based on a modified Anderson Level II classification system in the Albers Conical Equal Area projection.

The Landscape Mosaic (LM) is a tri-polar (ternary) classification model analyzing the relative contributions of any three land cover base types, i.e., agriculture and developed land to otherwise natural land, within a given local neighborhood. This classification scheme uses a moving window analysis to assign a LM label to the center pixel (grid cell) of the window based on the proportions of agriculture, natural land, and developed land within the window. The LM model has been used extensively to assess land cover composition at various scales to map and analyze landscape patterns [7–10], plant invasions [11], forest area changes [10, 12], and identification of candidate areas for ecological restoration [13]. It has been employed in European ecosystem assessments [14, 15], the assessment of green infrastructure of European cities [16], in the United States Resource Planning Act (RPA) forest assessments [17, 18] and was recommended as the primary indicator of landscape pattern by the "The H. John Heinz III Center for Science, Economics and the Environment" [19]. Such research and applications have emphasized the well-established concepts that landscape pattern is scale-dependent and that a multi-scale approach is required if the goal is to inform a variety of management and policy questions in a consistent way [20, 21].

In this paper, we describe improvements to the LM-product to better support environmental assessments. The motivations for the enhanced LM-product introduced here are to provide a user-friendly implementation in an open-source software package (LM-module), modify the ternary classification to improve thematic resolution of the LM, and to add new LM reporting features for statistical, temporal, and scale analyses. In this paper, we focus on the following three new aspects. First, how can we summarize LM status maps in a generic and comparable way, which retains the scale-dependent key information? Second, how can we quantify temporal changes of LM status maps? Third, we illustrate how the information content of the LM map can be easily represented in different ways to target different thematic monitoring interests. In addition, we outline how the enhanced LM analysis features could be beneficial in different application fields.

## Material

We used the United States National Land Cover Database (NLCD) to demonstrate the new LM-module enhancements in the free and open-source software packages GuidosToolbox (GTB) [22] for user-oriented interactive analysis as well as GuidosToolbox Workbench (GWB) [23] for unsupervised automated processing of large maps. The U.S. Geological Survey, in association with the Multi-Resolution Land Characteristics Consortium (https://mrlc.gov), provides land cover and time integrated land cover change (pixel class at time $t$ informs pixel class at time $t$-$1$) at 2-to-3-year intervals at 30m resolution with a 16-class legend based on a modified Anderson Level II classification system [24]. The latest release of NLCD (NLCD2021) includes land cover for 9 dates (2001, 2004, 2006, 2008, 2011, 2013, 2016, 2019, 2021) for the United States [6]. We used the terminal dates (2001 and 2021) for the applications that required land cover change data.

We chose NLCD because of their well-developed protocol for assessment of data quality [25, 26 and citations within], which follows well-established recommendations [27–29]. For example, all NLCD land cover accuracy assessments include estimates of map bias [27] since its establishment is an important component of accuracy assessment quantification [26 and

citations within]. Map bias is the error associated with area estimates based on pixel counts. Area estimation based on pixel counts is considered biased because a fundamental assumption of map accuracy assessment is that the higher quality reference data provide a truer area estimate.

Notwithstanding the well-established protocol for area estimation for mapped data, demonstration of LM-module enhancements was derived from the pixels in the map and thus map classification errors are inevitably included in the output present here, and any area estimates that could be derived from that output. Our rationale for acknowledging but not addressing map bias in this paper are both practical and well-established. Practically, reference classifications are based on a sample and not a census (i.e., map). As a result, the reference (classification) data are not appropriate input for the LM-module or other similar applications. Nearly all applications and assessments that require a map of land cover rely on the map itself as the source. Propagation of map bias into measurements derived from maps is quite difficult to avoid in most cases. On the other hand, if the bias is consistent between maps, then bias is probably less important when comparing measurements of LM on each map; ignoring the spatial aspect, and assuming the bias is constant over time, the change in a biased estimator is still an unbiased estimator of the change. Previous research has shown that bias in area-based map derivatives (e.g., patch size) was less than the imposed misclassification rate [30]. Such sensitivity analyses, however, are beyond the scope of this paper.

## Methods

As a ternary classification model, the LM metric requires a raster input map with three land cover base types. The choice of the three base types depends on the input data and research question, which is further detailed in the Discussion section. We used the assignment *A*, *N*, *D* (1-Agriculture, 2-Natural, 3-Developed) to demonstrate LM-module enhancements. Missing data (optional class value of 0) are simply ignored in the classification. Recoding of the 16-class NLCD map to the LM input map was conducted by recoding the NLCD classes 81, 82 to 1 (Agriculture), 21 through 24 to 3 (Developed), and all other NLCD classes to 2 (Natural) besides the NLCD class 0, which is reserved to mark pixels with nodata for our assessment, i.e., open ocean. The so-defined LM-input maps for the years 2001 and 2021 were then processed for the LM-metric. The LM was mapped using five analysis scales corresponding to five moving window sizes with side-lengths of 7, 13, 27, 81, 243 pixels corresponding to analysis scales of approximately 4, 15, 66, 591, 5314 hectares, respectively. Our choice of analysis scales was made to span a wide range of scales representing an approximately geometric progression of analysis scales. Other choices could be made in other circumstances; the implementation of LM-processing provides a variety of options and features that are summarized in a dedicated LM product sheet [31].

## The original 19-class LM map

As originally encoded, the ternary chart of the Landscape Mosaic was segmented into 19 mosaic classes, defined by applying threshold proportions of 0.1, 0.6, and 1.0 along each of the three axes (Fig 1A).

The mosaic names indicate the substantial presence (0.1 threshold), dominance (0.6), or uniqueness (1.0) of each land cover type. A lower-case letter (*a*-agriculture, *n*-natural, *d*-developed) in a mosaic class name indicates a relative proportion of at least 0.1 but less than 0.6, and an upper-case letter (*A*, *N*, *D*) indicates a proportion of at least 0.6 but less than 1.0. The LM-class is labeled with *AA*, *NN*, *DD* if the area covered by the moving window contains only one land cover class (ternary chart corners). Finally, a missing letter indicates that the

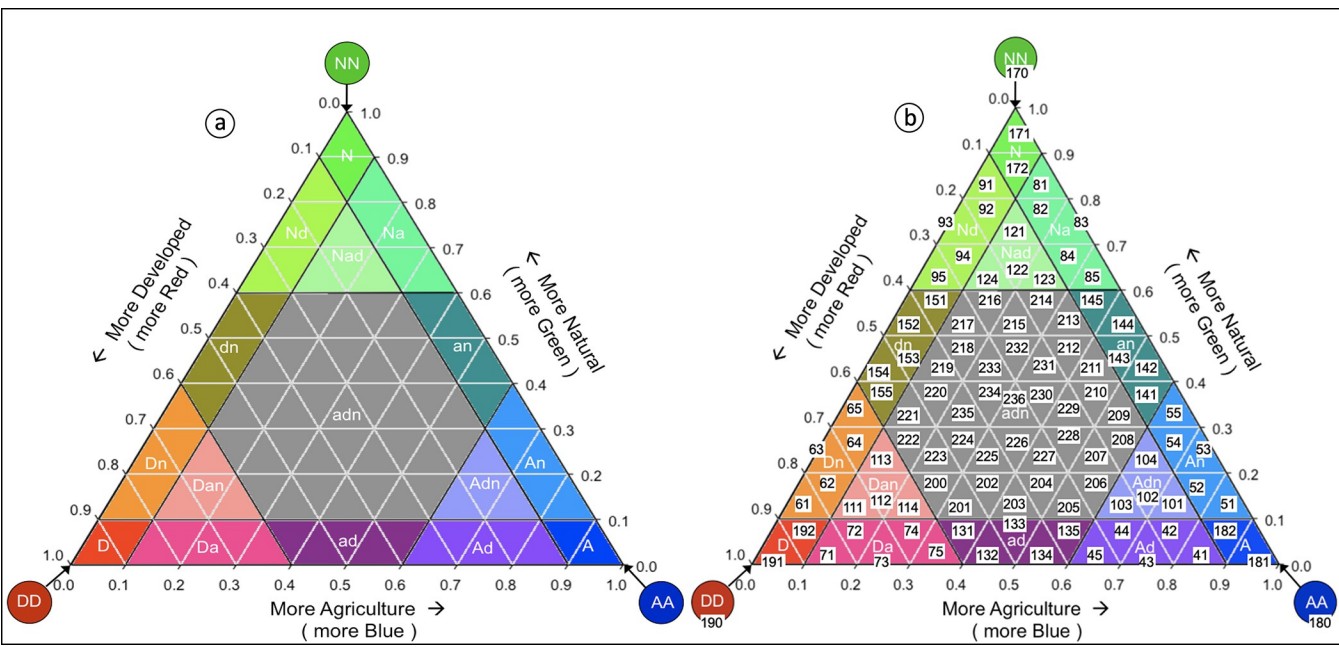

**Fig 1.** Legend for the original 19-class LM map (a), and the new enhanced 103-class LM map (b). Numbers in (b) are class codes.

respective land cover proportion was less than 0.1. With this notation, dominance is indicated by upper-case letters, an interface zone by a combination of upper- and lower-case letters, and mixture is indicated by lower-case letters only. We distinguish interface zone and mixture only to articulate features of the LM model, not to imply ecological or environmental distinction between the two zones in the LM model. When rendered as a map, the mosaic colors reflect the varying degree in color intensity with respect to the proportion of, developed (Red), natural (Green), and agriculture (Blue) land covers. The result of LM-processing as originally encoded is a spatially explicit map with up to 20 unique pixel values/colors for the 19 LM-classes and the nodata pixels (rendered in white).

## The enhanced LM-product

The enhanced LM-product output is composed of four parts: the original 19-class LM map (1); the more continuous 103-class LM map (2), the LM heatmap graphic (3), and the LM heatmap summary table in csv-format and a binary sav-format (4), used for change analysis.

## The new 103-class LM-map

A recent enhancement to LM-processing is the additional, more continuous output map with 103 mosaic classes: 100 for each sub-triangle within the ternary chart created by articulating a step size of 0.1 along each axis of the chart plus the 3 exclusive classes at the corners of the triangle (Fig 1B).

The new output map provides more thematic detail than before and adds more flexibility for interpretation because it does not impose the three somewhat arbitrary thresholds in the original product. In contrast to the original 19-class color-coded output map, the new additional 103-class output map applies a grayscale color table because visually interpreting a 103-color map can be difficult especially over small geographic extents. Pixel values in the 103-class map resemble their counterpart in the 19-class map but with more detail.

Corresponding 19- and 103-class pixel codes are listed in S1 File. Compared to the 19-class map, pixels on the 103-class map correspond to a more uniform and detailed partitioning of the ternary LM chart. This is particularly true for the intermixed sub-space labeled *adn*, color-coded in gray in the 19-class map (Fig 1A), for which all output map pixels have the same value of 16. The 103-class map discretizes *adn* into 37 classes, thus allowing for a more refined pixel-level assessment of the relative contributions of the three axes *A*, *N*, *D*.

## The new heatmap summary

Another new and additional output of LM-processing is the heatmap summary: the frequency distribution of the ternary class assignment (Fig 2). The heatmap consists of 103 occurrence classes, showing the relative pixel occurrence frequency in percent in each of the 103 classes.

For visual clarity, occurrence frequencies in the heatmap summary are given in rounded integer values. Precise occurrence frequency percentages for all 103 classes are provided in a separate csv-formatted spreadsheet and a binary sav-format for conducting heatmap change analysis. An empty class denotes no occurrence. A class with a number lists the occurrence frequency rounded to an integer value within the available range [0, 100]%. A class highlighted by a black background shows the maximum occurrence frequency. The sum of all 103 classes equals 100% or the occurrence frequency of all pixels in the map, when the ternary axes represent the entire map. In the example shown in Fig 2, the class *AA* has the 103-class code 180 (Fig 1B) and lists the occurrence frequency of 1, or in other words, 1% of the 103-class LM map pixels fall in the class 180, or class *AA*. Likewise, 2% of the 103-class LM map pixels fall in the class 170, or class *NN*, and most of the 103-class LM map pixels are in the class 101 with a contribution of at least 0.7 in agriculture, 0.1 in natural, and 0.1 in developed land cover, respectively.

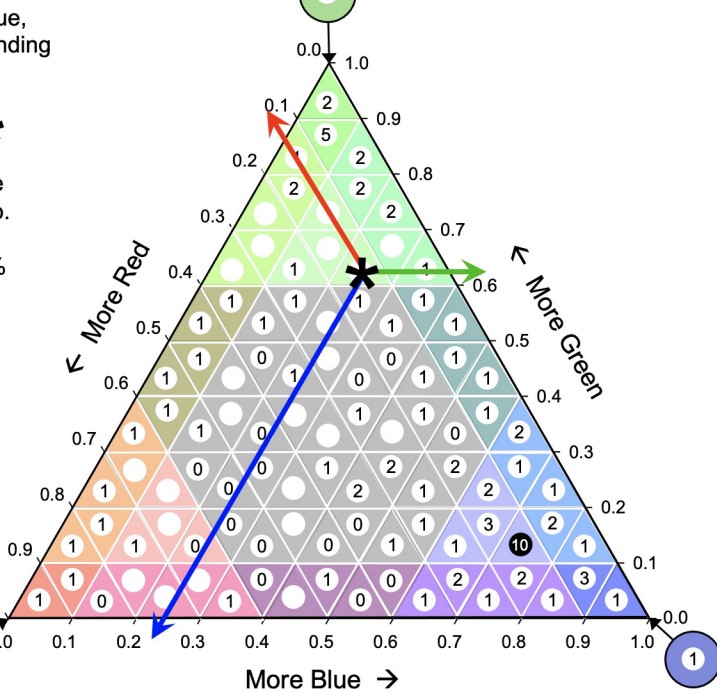

**Fig 2. Explanation of how to read the Landscape Mosaic pixel triplet values, denoting their relative contribution to blue, green, and red (here agriculture, natural, and developed) at a given location and for the local moving window area in use.**

## Results and discussion

### Reporting schemes

We illustrate how the information content of the more continuous 103-class LM map can be easily visualized in different ways to address different thematic monitoring interests. This is achieved by partitioning the LM domain space of the heatmap into a dedicated number of reporting categories matching a specific topic of interest. As with any classification, the selected thresholds are arbitrary and, depending on the specific topic of interest, they might be better represented with different thresholds or a different number of classes. For example, and depending on the application, the definition of natural or agriculture may be of different relevance or can vary significantly when considering different species in biodiversity assessments. However, the scope here is to illustrate generic concepts by providing suggestions for thematic reporting. A user can easily adapt any of the suggested reporting schemes to match user-specific definitions to adequately address individual aspects in the given application field.

In general, each pixel of the 103-class LM map shows the land cover composition, or the relative contribution of *A*, *N*, *D* within its local neighborhood (square moving window size). The various proportions of the land cover composition are color-coded in a predefined number of LM-classes and the feature space of the ternary LM-chart can be partitioned and color-coded into any thematic layer of interest, which is then visualized on the map. Fig 3 provides an example of Atlanta, of dimension 6000 x 6000 pixels (32,400 km$^2$), in 2021 for a moving window size of 13 pixels. Using the same 103-class map data, simply changing the zoning of the 103-class LM domain and assigning respective color-codes reveals various thematic features of interest.

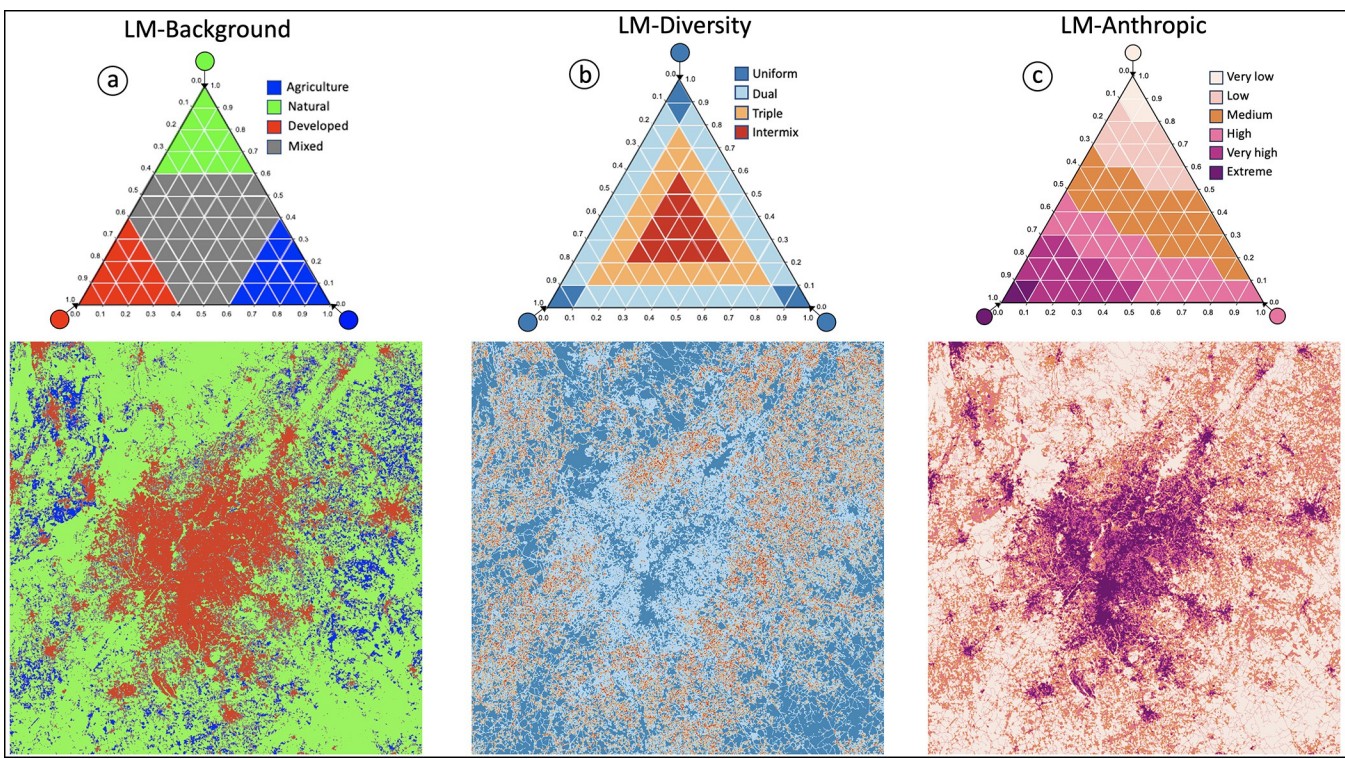

**Fig 3. LM around Atlanta for the year 2021 using a moving window size of 13 pixels.** Showing the 103-class LM map and applying different color tables to highlight dominant background land cover (a), degree of diversity (b), or anthropic intensity (c).

LM-Background (Fig 3A) shows a segmentation of the LM domain space into a simplified four-class color legend, highlighting pixels where the LM is either dominant in *A*, *N*, *D* or *mixed*. While LM-Background could have been derived from the original 19-class LM product, the point here is to illustrate how different zoning of the LM-domain can target different reporting questions.

LM-Diversity (Fig 3B) is an alternative four-class color legend showing the degree of heterogeneity in *A*, *N*, *D*. Here, the central part of the LM-Diversity domain space corresponds to pixels with almost equal contribution (< 33%) of all three contributors *A*, *N*, *D*. Pixels falling into this category have the highest degree in heterogeneity in *A*, *N*, *D*; they are named *intermix* and shown in red color. The next category of lesser but still high degree in heterogeneity, named *triple* pixels and shown in orange color, corresponds to pixels with predominant contribution of all three types of *A*, *N*, *D*. Pixels with predominant contribution of only two types of *A*, *N*, or *D* are named *dual* pixels and shown in light blue. Finally, pixels that are highly dominant (> 85%) in *A*, *N*, or *D* are called *uniform* and shown in dark blue. Map pixels falling into the category *uniform* exhibit a highly homogeneous neighborhood coverage in either *A*, *N*, or *D*. LM-Diversity is a reporting scheme exploiting the additional information available in the 103-class LM product only.

LM-Anthropic (Fig 3C) is designed to map the intensity of human activities caused by agriculture and particularly urban development in the surrounding neighborhood of a given location. We acknowledge that the two land cover categories agriculture and urban may have a different meaning and/or importance when interpreting land cover for specific applications in terms of their habitat quality, ecosystem services, or general value for ecosystems and biodiversity. These aspects can be addressed by the user by assigning a suitable weighting factor to each of the 103 LM-classes. To illustrate the idea in our example, we provide a generic weighting of the importance of A and D. Anthropic intensity or mapping the ternary information of anthropogenic activities into a normalized linear scale, is further illustrated in Fig 4.

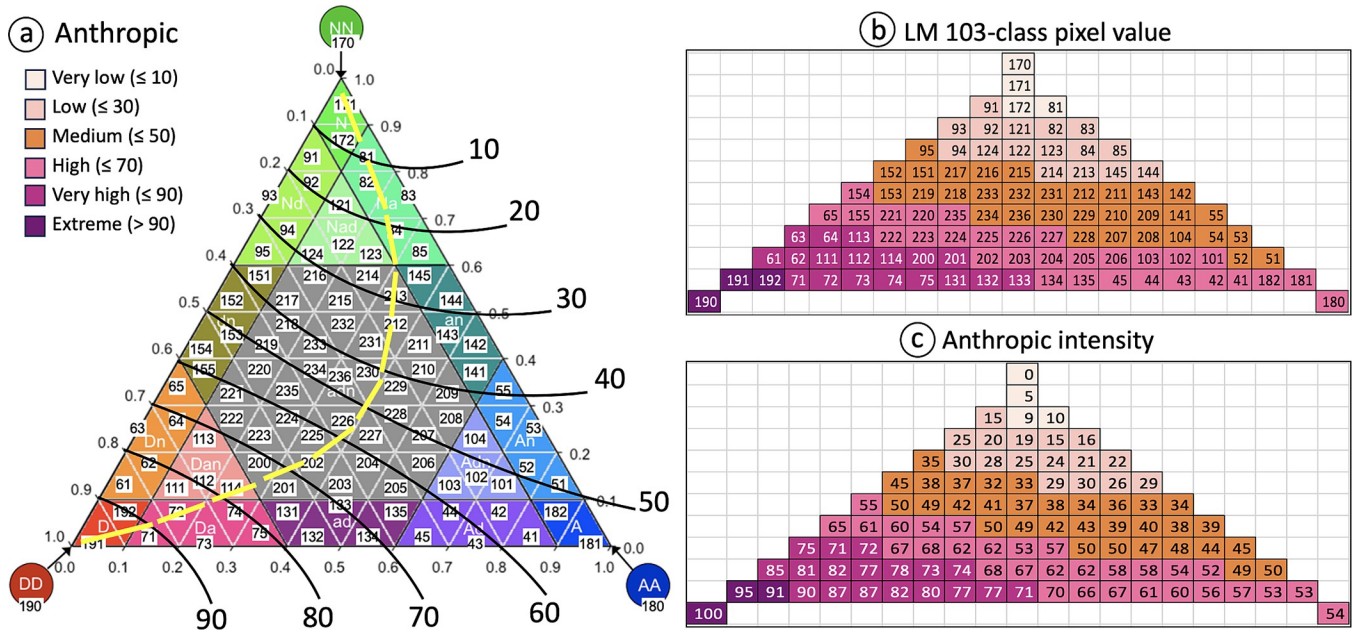

**Fig 4.** LM 103-class mapping to the degree of anthropic intensity (a); associated 103-class pixel color scheme (b); and respective anthropic intensity value (c).

Without any human activity, all land would arguably be natural and the anthropogenetic impact on natural ecosystems is further reviewed in the Discussion section. Hence, conceptually anthropic intensity is 0% for the class *NN* at the top of the LM triangle (LM 103-class pixel value 170) and 100% for the class *DD* at the bottom left corner of the LM triangle (LM 103-class pixel value 190). The black iso-lines in Fig 4A indicate increasing levels of anthropic intensity triggered by an increasing amount of agricultural and especially developed contributions. For example, the LM class 61, which is the third cell from the DD corner on the DD-NN axis in Fig 4A and 4B, has the anthropic intensity value of 85 (Fig 4C) resulting from interpolating the iso-lines in Fig 4A. The linear progression of anthropic intensity is symbolized by the yellow line in Fig 4A. For visual clarity, we grouped the full range of anthropic intensity [0, 100]% into six categories or color classes (legend in Fig 4A). Applying the LM-Anthropic color table will show the six degrees of anthropic intensity on the original 103-class LM map (Figs 3C and 4B). Alternatively, one can apply a recoding table (second and third column in S1 File), which will map the original 103-class pixel values (Fig 4B) to the respective anthropic intensity pixel values (Fig 4C) resulting in a map of normalized anthropic intensity with pixel values in [0, 100]%.

## Temporal change of the LM map product

Fig 5A shows the map of anthropic intensity over Atlanta for the years 2001 and 2021. The first visual impression suggests an increase of urban sprawl in the outskirts of Atlanta.

A more detailed analysis is possible by comparing the histogram of pixel values in anthropic intensity change for the two years. Here the visual impression of similar appearance (Fig 5A) is confirmed in the histogram with 73% of all pixels showing no change. To further investigate the changes, Fig 5B shows the histogram of pixels with changes only, which is 27% of the map region. In line with the urban growth around Atlanta, most change pixels exhibit a positive increase in anthropic intensity between 5% and up to 40% (Fig 5B). To visualize the locations

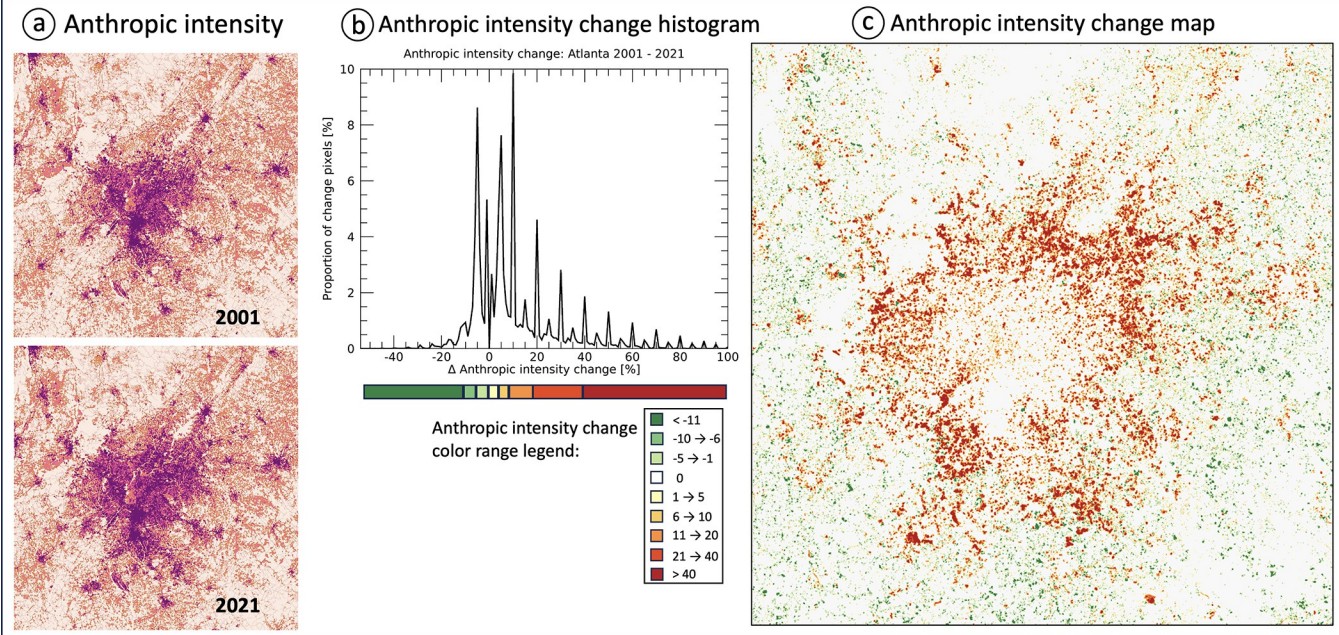

**Fig 5.** Anthropic intensity for Atlanta 2001 and 2021 (a); histogram of change pixels and anthropic intensity change color range legend (b); locations and degree of anthropic intensity change (c). Results based on 13-pixel window size (~15 ha).

and in particular the degree of change in anthropic intensity on the map, we define a custom color table by grouping the main sections of the histogram of change pixels (Fig 5B) and then apply this color table to the change map (Fig 5C), showing 1) a "halo" of anthropic intensity increase surrounding the urban core of Atlanta, and 2) localized areas of anthropic intensity decrease spread throughout the analyzed region.

## Temporal change of the LM heatmap product

Heatmap change analysis for two maps or two points in time can be conducted via the GTB–change–LM heatmap option, which computes the pairwise difference of each of the 103 heat-map entries. The result is a new delta heatmap where percentage decrease is shown in red numbers and increase in black numbers. The graphical delta heatmap is complemented with a respective delta heatmap table with precise change numbers in csv-format. Both the graphical and the tabular heatmap allow localizing the degree of change within each of the 103 classes of the LM domain space. Furthermore, we illustrate how the delta heatmap allows summarizing and quantifying temporal trends.

The heatmap for the year 2001 and 2021 is shown in Fig 6A and 6B. Fig 6C displays the delta heatmap with the difference in percentage within each of the 103 LM-classes. *Ax*, *Nx*, and *Dx* classes of the LM triangle are highlighted in blue, green, and orange color, and *mixed* is shown in gray color. The top-left chart in Fig 6C lists the LM-input proportion and change in agriculture, natural, and developed land cover of the input maps. The top-right chart in Fig 6C lists the delta heatmap summary for the four LM-Background classes *Ax*, *Nx*, *Dx*, and *mixed*. Percent changes are relatively small due to being confined to a very small subspace of the LM domain of 103 classes.

Two decades of urban growth around Atlanta become visible as an increase in urban land cover (LM-input *Dev* in Fig 6C) from 19.02% to 23.20%, or a relative increase of 21.96% (1353.35 km$^2$), within the area analyzed. Overall, the delta heatmap shows that urbanization has come at the expense of both natural and agricultural land cover. The additional value of the 103-class LM model is that it shows that urban gain is distributed across all levels of quantization of urban, agriculture, and natural. This level of insight is not possible with the 19-class model because of its coarser thematic resolution. Urban gain increased as the amount of natural decreased from 80% to 70% to 60% (i.e., positive values in black color along the NN-DD ternary axis, Fig 6C). The trend in this increase suggests that urbanization was more likely at higher levels of extant urbanization. The trend would be averaged into a single number in the 19-class model.

## LM change with analysis scale

The size of the moving window defines the analysis scale as the local neighborhood. Increasing the moving window size is equivalent to considering a larger local neighborhood, which has a blurring effect caused by the averaging of the increasing number of local pixel classes. Like comparing temporal trends (Fig 6), the delta heatmap may be used to quantify changes across analysis scales.

The map area is dominated by natural LM-classes (green colors) across all scales (Fig 7A). This is confirmed by the statistical summary table using the LM-Background reporting scheme (Fig 7B). To provide a more detailed change analysis summary, the *mixed* class of LM-Background is further subdivided into four classes of approximately equal extent. Three classes focusing on the three interface types (*dn2*, *an2*, and *ad2* in Fig 7B) and the remaining central *mixed* class *adn2*. Relative to the percentages at the smallest neighborhood size (7 x 7 pixels or 4 hectares), percentages of agriculture dominance (*Ax*), natural dominance (*Nx*) and urban

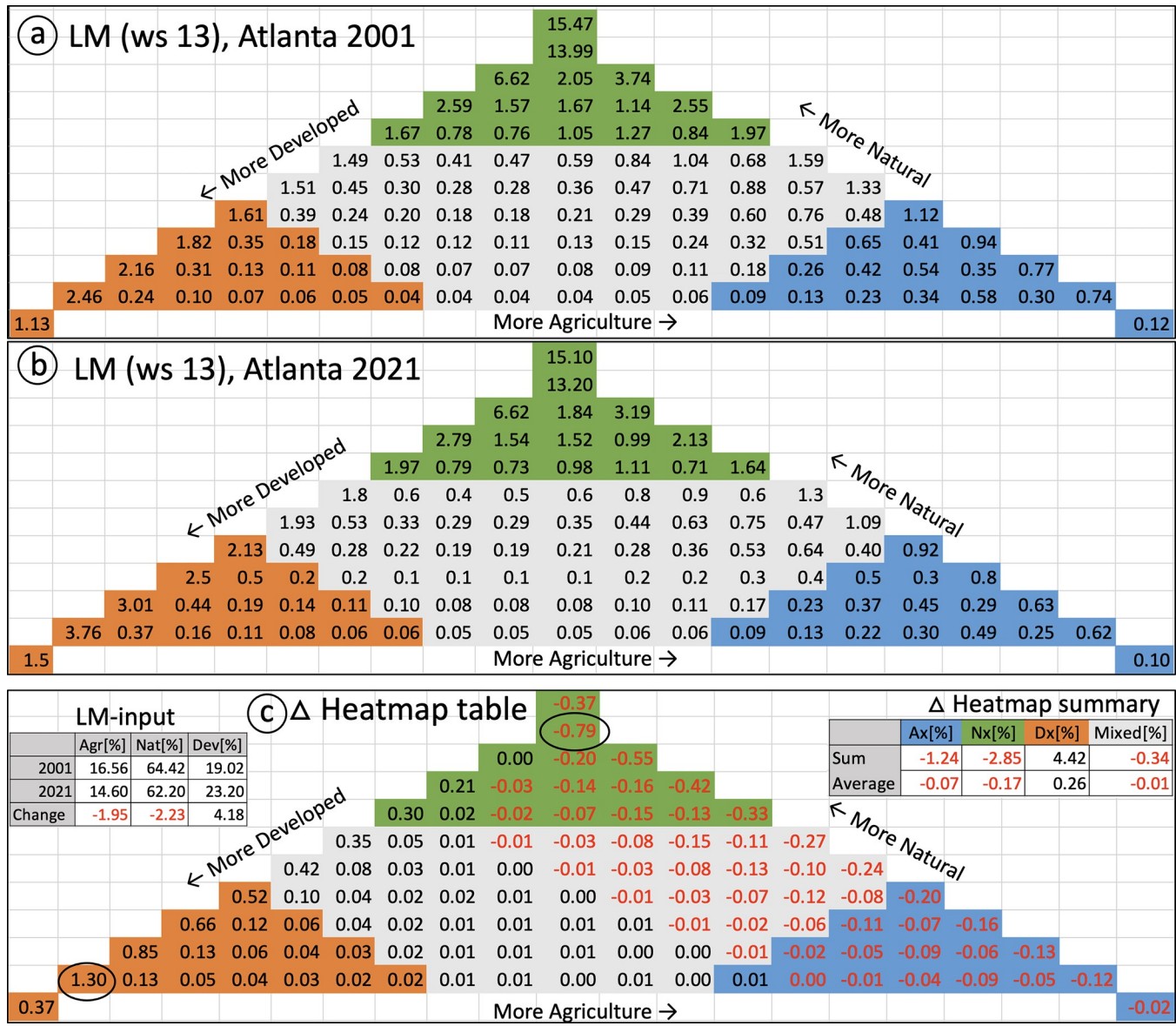

**Fig 6. Temporal change (2001–2021) of LM around Atlanta using a moving window size of 13 pixels.** LM heatmap for 2001 (a), LM heatmap for 2021 (b), and the delta heatmap showing change in LM-class percentages between the two LM heatmaps with the maximum increase and decrease highlighted in black circles (c).

dominance (*Dx*) at the largest scale (243 x 243 pixels or 5314 hectares) decline by 100%, 1%, 35%, respectively, but *mixed* increases by 92%. The summary table in Fig 7B further indicates that most of the increase in *mixed* is found in the class *dn2* and *an2*. Even more change details become available in the delta heatmap, summarizing the changes in each 103-class between the smallest and largest analysis scale (Fig 7C). While the class natural dominance (*Nx*) is constant across analysis scales, the 103-class LM product reveals strong fluctuations within the class *Nx*, including the overall highest increase and decrease. Similarly, we find that the increasing proportions in *mixed* are found at the transition zones from *Nx* to *dn2* and *an2*, while *ad2* and *adn2* remain virtually stable.

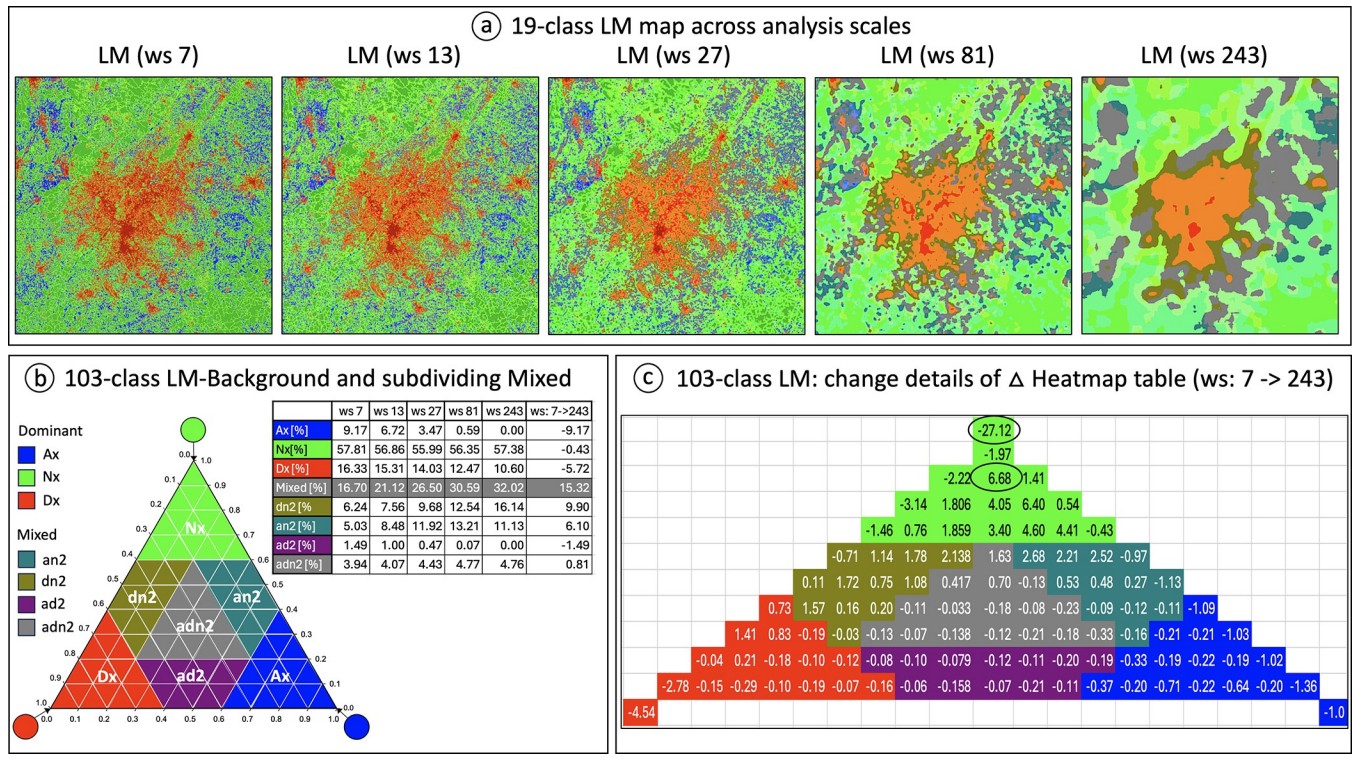

**Fig 7.** LM for the year 2021 around Atlanta: comparison of LM-classes across five analysis scales of moving window size (a); summary proportions per scale and difference in LM-Background and refined Mixed (b); delta heatmap between smallest and largest scale with the maximum increase and decrease highlighted in black circles (c).

These patterns suggest the following three conclusions: the spatial scale of agriculture dominance is less extensive than the spatial scale of natural land cover; urban areas are localized hotspots that are at the same time spatially pervasive, perhaps because of the extensiveness of roads [32]; and human activity tends to be locale but impacts also other non-local scales (e.g., the *mixed* percentage increases across scale). An alternative perspective is that the class natural could be understood as an absence of human impacts, whereases the other two classes (urban and agriculture) are a direct result of human impacts.

## Limitations

While the enhanced LM-product provides additional assessment and reporting options, the LM model as such is also subject to certain limitations. For example, no part of the globe is completely free of anthropic effects but that does not obviate the value of the LM model for analyzing a global map that fully characterized everything. Another obvious constraint of the LM model is that it cannot consider tele-coupling (long-distance, indirect effects, perhaps mediated through commerce), but again that does not obviate the value of the LM model for analyzing "local" spatial effects. Being an image analysis tool, the LM moving window approach inherits the bias and errors from the input map, and it is limited to the number and quality of the input land cover classes. Those again, depend on the satellite sensor, data processing, and definition of the land cover classes considering either, land use or actual land cover. Land cover pixel configuration within the moving window area also depends on the type of input map. Obviously, the type and quality of the input land cover map directly translates into the resulting LM-product and, as with any other image analysis assessment, the user

should be aware of these issues and select adequate input data. A LM-specific issue is the requirement to segment the land cover classes into three land cover base types for the LM chart. This decision may be difficult but should best match the user's expectations of for example which land cover classes should be considered as *natural* LM-classes. On the other hand, the flexibility of the LM model, its applicability to any suitable land cover dataset, and its variety of reporting options are a key strength of the approach.

While tasks involving validation, area, and proportion estimations in the context of the LM warrant further research, they are not within the scope of this study. Reference [27] offers guidance on approaches for these additional tasks, particularly in good practices for accuracy assessment and area estimation using land cover/use data.

## Conclusions

### LM summary

In this paper, we revisited the LM model by increasing the legend from 19 to 103 classes and introduced the heatmap to further support the enhanced thematic resolution. The LM heatmap is a normalized histogram distribution of the pixel values put into a thematic context. While the original 19-class model output may be easier to interpret and to communicate to a non-scientific community, the enhanced 103-class model introduced here includes the original 19-class model output but adds additional information on the landscape context.

We demonstrated the enhanced utility of the 103-class LM model using three examples — anthropic intensity, analysis scale, and temporal change analysis. The increase on the *D-N* axis in Fig 6C (e.g., 0.21, 0.30) indicates that urbanization occurred in locations that were nearly exclusively natural in 2001. Together these three positive values along the *D-N* axis, 0.00, 0.21, and 0.30, when expressed using the 2001 percentages as the basis, represent increases of 0.04%, 8%, and 18%, respectively in *D* increase from $\geq 10$, to $\geq 20$, to $\geq 30$ in 2001. This urban sprawl pattern would have been obscured in the 19-class model because it would have been averaged across all cell locations labeled *Dn*. At the same time, examination of the percentages along the *D-N* axis when $D \geq 60$ indicates that urban infilling is a predominant pattern. The largest increase among these classes (1.30%), and among all 103 LM classes occurs for *D*. The delta heatmap suggests that two common urbanization patterns [33], infilling and sprawl, occurred in the Atlanta metropolitan area between 2001 and 2021.

The anthropic intensity estimation, like the delta heatmap analysis, also relies on the 103-class LM model. The application presented demonstrates an analysis weighting urban over agriculture that could easily be reconstructed using other weighting criteria. An analysis weighting agriculture rather than urban in rural areas might be useful for examining impacts on natural resources in rural settings.

The scale analysis showed how much each of the LM-Background classes decreased and that only the central class *mixed* increased with increasing analysis scale. While this behavior may be specific for the land cover extent analyzed, the scale-specific LM change behavior is an additional information source that may be of interest when designing or evaluating the impact of local to regional landscape planning programs. To enable a suitable LM analysis, GTB provides a dedicated interface allowing the user to insert any appropriate custom analysis scale.

### Potential applications

Content versus context and interface zones are important themes in ecology, environmental management, and sustainability. What surrounds a particular feature (context) can be as important to its long-term sustainability as its content [34]. Ecological dynamics at the interface zone, a content-context boundary, for example, has important implications for

management of natural resources. Content as related to context, interface zones, and similar terms are specific expressions of ecological boundaries. The LM is a tool that supports study of ecological boundary effects including many (but not all) of its expressions and conceptualizations [35]. We offer a few examples of potential applications.

Content versus context is an issue which many protected areas must address [34–36]. Urban and agricultural land cover surrounding a protected area may pose threats that cannot be addressed simply by ecosystem management within the protected area. The LM provides that context. GIS software can be used to embed protected areas within the LM to assess the context within which the protected area resides. As demonstrated here with the temporal dimension of NLCD, temporal change surrounding a protected area may be an important aspect of its management. Recent research has shown that landscape context as defined by LM was an important determinant of forest loss rates [10, 12].

The LM model is a scale-dependent (window size) classification of interface zones in that each pixel is classified according to a normalized distribution of three surrounding features (e.g., agriculture, urban, natural). The wildland urban interface (WUI), an area with intermixed human infrastructure and natural lands, is an extensively studied boundary popularized in the United States [37–39] that is now used globally [40–42]. Identification of the WUI relies on census and land cover data. The LM model can be used to identify the WUI without reliance on census data, providing an additional means of WUI identification where census data may be difficult to acquire. Reference [43] examined the effect of variation in the WUI definition on identification of the interface itself. Use of the LM model to identify the WUI adds an additional method to incorporate, one that can be used to introduce scale-dependence into its identification. More broadly, the LM model identifies agriculture-developed, agricultural-natural, and agriculture-developed-natural interface zones.

Context is an active area of research in ecology. In agricultural settings, landscape context influences the diversity and abundance of pollinators [44, 45], host-parasite relationships [46], and the magnitude of adverse impact of pesticides on bee populations [47]. In other settings, landscape context has been influential in habitat restoration outcomes [48], a determinant of forest edge characteristics [49], useful for improving forest density estimates derived from LIDAR data [50], and an effective predictor of avian species occurrence [51]. The literature on landscape context is very extensive, perhaps even overwhelming to fully consider. We highlighted a few examples here to provide an admittedly uncalibrated overview of the potential areas of application of the LM model.

The landscape context provided by LM also has an intuitive application to identification of areas for ecological restoration. LM output is likely to be a useful starting point for identification of candidate areas for ecological restoration, for example by locating dominant natural land cover. The horizontal lines extending from the base of the heatmap to the *NN* apex represent increasing proportions of natural land cover. Because the cost of restoration generally increases with the amount of area to be restored [52], restoration undertaken in the intermix (three lower case letters) or interface zones (i.e., *Dx, Nx, Ax*) with higher proportions of natural land cover may harbor areas where both cost-efficiency and restoration benefits could be maximized [13]. Additionally, metrics that describe mosaics of land system types at the landscape level are key to assessing ecosystem conditions using international standards, such as the UN's System of Environmental-Economic Accounting–Ecosystem Accounting (SEEA-EA, https://seea.un.org/ecosystem-accounting) [15]. Metrics like the LM described in this paper quantify characteristics at the landscape scale that affect the local condition of ecosystems and can be attributed to specific pressures, such as intensive agriculture or urban expansion.

An additional LM model asset is its flexibility. The model apices, perhaps most notably applied to percentages of sand, silt, and clay for soil classification, are user-specified (see

Methods). We specified three intuitive themes that are common to most land cover maps (agriculture, urban, natural). For study of vegetation dynamics in the sub-humid to semi-arid climates, the apices could be defined as shrubland, grassland, and forest to gain an understanding of spatial patterns, scale dependencies, and interaction among these landscape features.

## Supporting information

**S1 File. The spreadsheet file provides a list of corresponding LM 19-class, LM 103-class and anthropic intensity pixel values.**
(XLSX)

**S2 File. This archive contains example LM-output files for the Atlanta 2001 map.**
(ZIP)

**S3 File. This archive contains example LM-output files for the Atlanta 2021 map.**
(ZIP)

## Acknowledgments

The views and opinions expressed in this article are those of the authors and do not necessarily reflect the official position of the European Commission. This work was performed in the context of a research agreement between the USDA Forest Service (Memorandum of Understanding #14-MU-11330110-001) and the European Commission Joint Research Centre (Collaborative Research Arrangement #33385).

## Author Contributions

**Conceptualization:** Peter Vogt, James Wickham, José Ignacio Barredo, Kurt Riitters.

**Data curation:** Peter Vogt.

**Formal analysis:** Peter Vogt.

**Methodology:** Peter Vogt, James Wickham, José Ignacio Barredo, Kurt Riitters.

**Software:** Peter Vogt.

**Writing – original draft:** Peter Vogt.

**Writing – review & editing:** James Wickham, José Ignacio Barredo, Kurt Riitters.

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
