## [Decision Letter · Decision Letter 0]

19 Feb 2024

PONE-D-23-39102Revisiting the Landscape Mosaic modelPLOS ONE

Dear Dr. Vogt,

Thank you for submitting your manuscript to PLOS ONE. After careful consideration, we feel that it has merit but does not fully meet PLOS ONE’s publication criteria as it currently stands. Therefore, we invite you to submit a revised version of the manuscript that addresses the points raised during the review process.

**The two reviewers provided useful guidance related to this study that the authors should carefully consider during revision. Careful attention in revision with thoughtful reply will be needed to assuage reviewer 2's concern regarding statistical rigor of this study, which if un-addressed, could potentially lead to a rejection. I encourage the authors to submit major revisions to the article and look forward to seeing the revised version.**

We look forward to receiving your revised manuscript.

Kind regards,

Kristofer Lasko, PhD

Academic Editor

PLOS ONE

Journal Requirements:

Whilst you may use any professional scientific editing service of your choice, PLOS has partnered with both American Journal Experts (AJE) and Editage to provide discounted services to PLOS authors. Both organizations have experience helping authors meet PLOS guidelines and can provide language editing, translation, manuscript formatting, and figure formatting to ensure your manuscript meets our submission guidelines. To take advantage of our partnership with AJE, visit the AJE website (http://aje.com/go/plos) for a 15% discount off AJE services. To take advantage of our partnership with Editage, visit the Editage website (www.editage.com) and enter referral code PLOSEDIT for a 15% discount off Editage services. If the PLOS editorial team finds any language issues in text that either AJE or Editage has edited, the service provider will re-edit the text for free.

Reviewers' comments:

Reviewer's Responses to Questions

**Comments to the Author**

1. Is the manuscript technically sound, and do the data support the conclusions?

Reviewer #1: Yes

Reviewer #2: No

2. Has the statistical analysis been performed appropriately and rigorously? 

Reviewer #1: Yes

Reviewer #2: No

3. Have the authors made all data underlying the findings in their manuscript fully available?

Reviewer #1: No

Reviewer #2: Yes

4. Is the manuscript presented in an intelligible fashion and written in standard English?

Reviewer #1: Yes

Reviewer #2: Yes

5. Review Comments to the Author

Reviewer #1: **General remarks**:

The manuscript introduces an improvement to the existing Landscape Mosaic Model and presents an interesting and promising approach to quantify landscape characteristics. However, the manuscript could be improved at several points.

First, it is currently unclear to the reader which parts relate to the already existing approach and which parts are newly introduced in the context of this research. This mainly refers to the Methods section.

Second, the manuscript would benefit from a brief but clear introduction of the already existing approach and a following clear description of the improvements suggested here. Generally, the exact steps of the improvements suggested in this manuscript are not fully clear and the Methods section should be improved in terms of clarity.

Third, during the Introduction section, but also during the Discussion section it would be interesting to compare the here introduced methods to already existing approaches to quantify landscape characteristics. Also, the introduction is missing information on why the existing approach needs to be improved.

Fourth, the manuscript could be improved by revisiting the Methods and Results section - ensuring a clear separation between the two. Currently especially in the Results section there are several paragraphs which would be helpful in the Methods.

Fifth and last, the manuscript includes quite a few figures and might be easier to approach by readers if some figures are moved to the Supplementary materials.

**Specific remarks**:

Line 37: Throughout the manuscript, but especially in the Introduction section, land-cover and land-use are used exchangeable, however, the terms describe slightly different but related aspects of a landscape.

Line 44: I agree that composition is a crucial point, however, what about configuration?

Line 47: It might be helpful, especially for readers not familiar with the topic, to briefly explain how remote sensed imagery gets translated in the pixel/cell based LULC maps.

Line 58 – 65: This section of the manuscript could be shortened and instead readers might benefit more from an improved explanation of landscape mosaic model.

Line 63: There is also plenty of literature proposing other methods to quantify landscape structure and patterns.

Line 67: So far, no scale-dependent aspects were introduced at all. However, this would be helpful, especially since different scales are explored later.

Line 84: What if the landscape cannot be converted into a 3-classes LULC map? This is a strong, simplifying assumption, given the complexity of existing landscapes. Thus, this requires some additional justification.

Line 95: Are the sizes of the moving windows based on typical moving window sizes used in previous studies? What is the basis for this selection?

Line 99-116 and Line 117-129: This paragraph needs some improvement. Currently, it is hard to understand what part of the existing landscape mosaic model is, and what was improved in this manuscript.

Line 177 – 182: If the extent of the landscape and by that the overall heterogeneity of the landscape has such a big influence of the results, this should be discussed more critically.

Line 195 – 200: It is unclear what this means. First, scale itself is not extensive (or not), rather different characteristics of the landscape might be more quantifiable at different

scales. The three conclusions are very difficult to understand.

Line 273: What does highly dominant or predominantly mean. What are the exact thresholds to classify this?

Line 278: The anthropic context combines two classes, namely agriculture and urban, correct? These two classes have very different characteristics in terms of their habitats, ecosystem services or general importance and value for ecosystems and e.g., biodiversity. For example, species experience an agricultural landscape probably very different than a developed city, even though both are created by humans.

Line 287: Even in natural landscapes the anthropic intensity is probably not 0% - in most regions of the world nature was affected at one point or another by humans. For example, especially in central Europe, but also many other parts of the world, even “natural” forests were shaped by human interventions.

Line 327 (but also others): In general, all presented relative changes of the landscape seem to be rather small (below 10% in many cases). A better contextualization of these magnitudes would be helpful.

Line 348: Configuration seems to be important as well, which gets not captured by the presented approach as far as I understand it.

Line 358: There are many other tools that help to inform where protected areas should be located. What are the advantages of the approach presented here?

Line 368: Identify the WUI using the approach presented in this manuscript will only be possible if values can be connected to known WUI areas (so basically a ground-truthing or training dataset which puts the WUI in relation to outputs of the approach).

Line 376: This sentence seems out of context.

Reviewer #2: This article provides an interesting study of landscape context using the landscape mosaicking technique based on the time-series of NLCD land cover classification maps. The study offers an enhanced assessment of land cover class changes and distributions over various sizes of neighborhoods for three types of land use – agriculture, natural, and urban.

In my opinion, the main weaknesses of this study are the statistical principles used for data aggregation and analysis, and the absence of validation. The proportions of pixels in the neighborhood were estimated using generic pixel counting without taking into account omission and commission errors. This approach undermines the main strength of the study (the use of a 103-class scheme that enhances thematic context) by introducing high level of errors in the proportion-based taxonomy. Moreover, the study does not provide any validation of the produced products or any assessment for the estimated statistics.

More detailed comments:

1. In the abstract, please add the institution that provides NLCD to be more specific.

2. L52: Please specify the projection of the maps

3. I think the introduction should be extended. Authors should provide more details on the impact of land cover/land use on the environment, the segmentation of land cover, integration into monitoring systems at the country or regional level, and policy-making. Most importantly, they should focus on the Landscape Mosaic concept and provide specific examples and benefits of its use. The introduction lacks a literature review.

4. The images provided by the authors in the article have a resolution that is too low.

5. I recommend separating the Materials and Methods section into two separate sections. One section should describe the used dataset, while the other should outline the processing and analysis methods.

6. L196-200, please provide quantification for the stated facts.

7. In the section “change in land cover context” please provide all numbers in the same numerical formats and with the same precision

8. It is bad practice to estimate proportions or areas for land cover classes, and especially land cover changes, using generic pixel counting. The maps used have omission and commission errors, and aggregating the time-series of the land cover maps into classes based on the percentage of pixels in the neighborhood can lead to a high level of errors and incorrect statistical analysis. The 30 m remote sensing-based land cover classification maps have numerous problems (mixed pixels, image mis-registration, classification model errors) that do not allow for unbiased estimation of areas/proportions [1]. Thus, authors should either integrate statistical information from the maps' validation reports into their analysis or use a sample-based stratified approach for area estimation [2,3].

References:

1. Olofsson, P. et al. Mitigating the effects of omission errors on area and area change estimates. Remote Sens. Environ. 236, 111492 (2020).

2. Olofsson, P. et al. Good practices for estimating area and assessing accuracy of land change. Remote Sens. Environ. 148, 42–57 (2014).

2. Cochran, William Gemmell. Sampling techniques. john wiley & sons, 1977.

9. No uncertainty analysis, confidence intervals, p-values, or accuracies were provided in this study.

10. L351-355 please provide more details about specific of the generated information use

11. The authors should also provide more information on the aggregation of NLCD classes into the three classes used. Especially important details should be provided regarding the differences between land cover and land use. For instance, an agriculture field can remain unsaved for several years with extensive cover of natural vegetation before the season when it will be planted. As a result, land cover maps can show the change from cropland to natural vegetation; however, in terms of land use and your taxonomy, it is still considered agriculture.

12. Authors should provide discussion of the weaknesses of the proposed approach and problems of the generated data.

6. PLOS authors have the option to publish the peer review history of their article (what does this mean?). If published, this will include your full peer review and any attached files.

Reviewer #1: No

Reviewer #2: No

---

## [Author Response · Author response to Decision Letter 0]

28 Mar 2024

Reviewer #1: **General remarks**:

The manuscript introduces an improvement to the existing Landscape Mosaic Model and presents an interesting and promising approach to quantify landscape characteristics. However, the manuscript could be improved at several points.

Response: We appreciate the reviewer's clarity and specificity on actions to improve the manuscript.

First, it is currently unclear to the reader which parts relate to the already existing approach and which parts are newly introduced in the context of this research. This mainly refers to the Methods section.

Response: Excellent point. The Methods section now opens with a clear distinction of past LM model capabilities and the new capabilities introduced in this paper. This is now also described in the Discussion paragraph. 

Second, the manuscript would benefit from a brief but clear introduction of the already existing approach and a following clear description of the improvements suggested here. Generally, the exact steps of the improvements suggested in this manuscript are not fully clear and the Methods section should be improved in terms of clarity.

Response: Thank you, we have rewritten the sections Introduction and Methods accordingly to better distinguish between the existing scheme and the new features that were introduced in our manuscript. 

Third, during the Introduction section, but also during the Discussion section it would be interesting to compare the here introduced methods to already existing approaches to quantify landscape characteristics. Also, the introduction is missing information on why the existing approach needs to be improved.

Response: Thank you for the comment. We have revised the introduction on the rationale for adding new capability to the LM model. We agree that a comparison to other landscape characteristic methods would be interesting, but this should be better addressed in a different paper, while our paper is focused on enhancements to the existing LM model. 

Fourth, the manuscript could be improved by revisiting the Methods and Results section - ensuring a clear separation between the two. Currently especially in the Results section there are several paragraphs which would be helpful in the Methods.

Response: Thank you for the comment. We have revised the methods section to bring more clarity. 

Fifth and last, the manuscript includes quite a few figures and might be easier to approach by readers if some figures are moved to the Supplementary materials.

Response: We now constrain the analysis on Atlanta only. In addition, two figures on scale were merged into one figure. We believe that the current number of figures are needed to explain the features of the enhanced LM product, and to illustrate the new features on reporting, change, and scale. 

**Specific remarks**:

Line 37: Throughout the manuscript, but especially in the Introduction section, land-cover and land-use are used exchangeable, however, the terms describe slightly different but related aspects of a landscape.

Response: Thank you for spotting this inconsistency. We have revised the text accordingly using the term land cover only and providing a footnote to a book section explaining differences between land cover and land use. 

Line 44: I agree that composition is a crucial point, however, what about configuration?

Response: It is well-established (e.g., Gustafson 1998) that composition and configuration are the fundamental aspects of landscape pattern. It is also well-established that some aspects of configuration can be inferred by measuring only composition (e.g., review by Riitters 2019). In the present case, we can infer juxtaposition (one aspect of configuration) to some extent by measuring the composition in a window. It is possible to perform a fuller analysis of configuration, for example by examining the spatial adjacencies of agriculture, developed, and natural pixels within a window, but then the LM model would not apply. 

Line 47: It might be helpful, especially for readers not familiar with the topic, to briefly explain how remote sensed imagery gets translated in the pixel/cell based LULC maps.

Response: We agree that it is important to inform readership as fully as possible. However, land cover classification is so commonplace now that the reader might find such a discussion of the topic superfluous, even distracting. To reconcile, we included the URL https://www.sciencedirect.com/topics/earth-and-planetary-sciences/land-use-classification. Open access links to a wide array of topics by Science Direct seems a good solution to us. Additional information was added in the Data section. 

Line 58 – 65: This section of the manuscript could be shortened and instead readers might benefit more from an improved explanation of landscape mosaic model.

Response: We have revised the sentence beginning on Line 56 (original submission) to “The tri-polar (ternary) classification scheme uses a moving window analysis to assign a LM label to the center pixel of the window based on the proportions of agriculture, natural land, and developed land within the window.” We have added an improved explanation of the landscape mosaic model in the new Methods section.

Line 63: There is also plenty of literature proposing other methods to quantify landscape structure and patterns.

Response: We agree. However, the objective of the paper is to document updates to the existing Landscape Mosaic model as well as its implementation into a widely used software package (GuidosToolbox). Despite it, we have included in the introduction a review of methods to quantify different landscape metrics citing key references (line 64-73). 

Line 67: So far, no scale-dependent aspects were introduced at all. However, this would be helpful, especially since different scales are explored later.

Response: Thank you for this suggestion. In the revised Introduction section, we have now added an explanation and reference outlining why scale assessments are important, hence justifying the need to address this topic in our manuscript. 

Line 84: What if the landscape cannot be converted into a 3-classes LULC map? This is a strong, simplifying assumption, given the complexity of existing landscapes. Thus, this requires some additional justification.

Response: A ternary classification requires that the sum of the percentages on the three axes is 100%. One feature of the LM implementation is that any of the original classes which do not fit into the 3 primary classes can be coded as “missing data” and thus are ignored in the computation of the percentages for a given window. In that way, any assumptions can be made by the user, and do not require justification here. For example, some of the earlier applications of the LM model set “water” to “missing data” in order to not dilute the signal of development along coastlines. Our paper illustrates the generic approach; the supporting user guide outlines options for re-coding. We also added a paragraph of limitations of the approach at the end of the Discussion section.

Line 95: Are the sizes of the moving windows based on typical moving window sizes used in previous studies? What is the basis for this selection?

Response: Window sizes are user-specified. We have revised the text to make this point more clear. Various scales are needed to investigate the LM-results across scales and to justify our choices we added the sentence “Our choice of analysis scales was made to span a wide range of scales representing an approximately geometric progression of analysis scales.”

Line 99-116 and Line 117-129: This paragraph needs some improvement. Currently, it is hard to understand what part of the existing landscape mosaic model is, and what was improved in this manuscript.

Response: We have revised the new Methods section to improve clarity and clearly define the novelty of the new approach in this paper.

Line 177 – 182: If the extent of the landscape and by that the overall heterogeneity of the landscape has such a big influence of the results, this should be discussed more critically.

Response: We have deeply revised the Introduction, Materials, and in particular Methods section to better explain the new enhancements, which include an analysis at various scales. The latter is important to evaluate various amounts of landscape extent and its local heterogeneity, which is a prime feature of the LM analysis. 

Line 195 – 200: It is unclear what this means. First, scale itself is not extensive (or not), rather different characteristics of the landscape might be more quantifiable at different scales. The three conclusions are very difficult to understand.

Response: The reviewer is correct that landscape patterns are expressed (and perceived) differently at different scales; landscape patterns are scale-contingent. A single-scale demonstration would simplify the manuscript, but a multi-scale demonstration informs readers who may not be interested in the single scale that we arbitrarily select.

Line 273: What does highly dominant or predominantly mean. What are the exact thresholds to classify this?

Response: We have revised the text adding threshold values to make this concept more clear.

Line 278: The anthropic context combines two classes, namely agriculture and urban, correct? These two classes have very different characteristics in terms of their habitats, ecosystem services or general importance and value for ecosystems and e.g., biodiversity. For example, species experience an agricultural landscape probably very different than a developed city, even though both are created by humans.

Response: We revised the text to accommodate this valid observation and explain that we illustrate the idea of a generic concept here. The reviewer is correct that agriculture and urban areas are likely to have very different ecological implications. One feature of the LM model is that both anthropic stresses are quantified and mapped using a consistent method. As we illustrate, the weighting of individual partitions of the LM with respect to anthropic context can be easily amended and redefined by the user in the user-specific context.

Line 287: Even in natural landscapes the anthropic intensity is probably not 0% - in most regions of the world nature was affected at one point or another by humans. For example, especially in central Europe, but also many other parts of the world, even “natural” forests were shaped by human interventions.

Response: Thank you for this important observation, which is correct. No part of the globe is completely free of anthropic effects. That does not obviate the value of the LM model for analyzing a global map that fully characterized everything. One obvious constraint of the LM model is that it cannot take into account tele-coupling (long-distance, indirect effects, perhaps mediated through commerce), but again that does not obviate the value of the LM model for analyzing “local” spatial effects. Moreover, and due to the lack of any additional related information and in particular from a practical perspective, we are confined to the information provided by the land cover map. The methodology we describe is a post-processing methodology, which applies data analysis of the underlying map data. As with any other image analysis, we believe that the user is aware that our methodology is constrained to the map-inherent information and its quality. However, the user could define ‘natural’ in any other way and then use the same LM concept with this new definition of ‘natural’. 

Line 327 (but also others): In general, all presented relative changes of the landscape seem to be rather small (below 10% in many cases). A better contextualization of these magnitudes would be helpful.

Response: We have added a more detailed description to explain why some changes are relatively small, in particular the changes within each of the 103 LM-classes, which are explained by being confined to a very small subspace of the LM chart (line 320). 

Line 348: Configuration seems to be important as well, which gets not captured by the presented approach as far as I understand it.

Response: Please refer to the response to line 44, above. 

Line 358: There are many other tools that help to inform where protected areas should be located. What are the advantages of the approach presented here?

Response: Thank you for this observation. This is true and it could be the topic of a future paper. In our current paper we describe one example application and comparing it to protected area assessment schemes is beyond the scope of this paper. Our paper’s topic is on enhancements of the existing LM. The designation of protected areas can be seen as a post-hoc application of the results (maps) delivered in this study. Aspects such as natural areas scarcity in peri-urban areas and the services they could provide to citizens in urban areas such as recreation opportunities, clean air, climate regulation, are important topics. However, this is beyond the scope of this paper. Another interesting application of the LM model could be to examine the scale(s) at which a given protected area is actually “protected” For example, a small protected area could be free of anthropic stress in a small window, yet contained within a larger landscape (window) that was completely urbanized. In an earlier paper (Riitters et al 2018, https://doi.org/10.3390/f9110723) we addressed a similar question of the efficacy of “protection” with regard to invasive plants, using a simplified version of the LM model. A decision about where to locate a new protected area could consider this aspect, which is also related to the size of the proposed protected area.

Line 368: Identify the WUI using the approach presented in this manuscript will only be possible if values can be connected to known WUI areas (so basically a ground-truthing or training dataset which puts the WUI in relation to outputs of the approach).

Response: Yes, this again could be material for an additional paper. However, it would be distracting here to go into more detail on WUI since the focus of our paper is on the enhanced LM product. There is not a general definition of “WUI” based on land cover alone. In the US, the “official” national definition is articulated at the census “block” level, and combines forest cover data with housing density data, and ignores agriculture. We think the LM could be a starting point for another WUI definition that also incorporates agriculture (WUAI). 

Line 376: This sentence seems out of context.

Response: Thank you, we agree and this sentence has been removed.

Reviewer #2

This article provides an interesting study of landscape context using the landscape mosaicking technique based on the time-series of NLCD land cover classification maps. The study offers an enhanced assessment of land cover class changes and distributions over various sizes of neighborhoods for three types of land use – agriculture, natural, and urban.

Response: We appreciate the reviewer’s thoughtful review of the paper.

In my opinion, the main weaknesses of this study are the statistical principles used for data aggregation and analysis, and the absence of validation. The proportions of pixels in the neighborhood were estimated using generic pixel counting without taking into account omission and commission errors. This approach undermines the main strength of the study (the use of a 103-class scheme that enhances thematic context) by introducing high level of errors in the proportion-based taxonomy. Moreover, the study does not provide any validation of the produced products or any assessment for the estimated statistics.

Response: We understand the reviewer’s appreciation of the importance of statistically rigorous accuracy assessment of land cover data. One of the reasons NLCD was chosen as the database to demonstrate the new enhanced LM-product was the high quality of the NLCD data, demonstrated by previous statistically rigorous accuracy assessments of these data. NLCD accuracy assessments were built on the good practice recommendations in Stehman and Czaplewski (1998) and further recommendations outlined in Olofsson et al. (2014) for all NLCD accuracy assessments published a

---

## [Decision Letter · Decision Letter 1]

29 Apr 2024

PONE-D-23-39102R1Revisiting the Landscape Mosaic modelPLOS ONE

Dear Dr. Vogt,

Thank you for submitting your manuscript to PLOS ONE. After careful consideration, we feel that it has merit but does not fully meet PLOS ONE’s publication criteria as it currently stands. Therefore, we invite you to submit a revised version of the manuscript that addresses the points raised during the review process. 

**One reviewer has requested several minor edits to the manuscript. Please revise and re-submit accordingly.**

We look forward to receiving your revised manuscript.

Kind regards,

Kristofer Lasko, PhD

Academic Editor

PLOS ONE

Journal Requirements:

Reviewers' comments:

Reviewer's Responses to Questions

**Comments to the Author**

1. If the authors have adequately addressed your comments raised in a previous round of review and you feel that this manuscript is now acceptable for publication, you may indicate that here to bypass the “Comments to the Author” section, enter your conflict of interest statement in the “Confidential to Editor” section, and submit your "Accept" recommendation.

Reviewer #1: (No Response)

Reviewer #2: All comments have been addressed

2. Is the manuscript technically sound, and do the data support the conclusions?

Reviewer #1: Yes

Reviewer #2: Yes

3. Has the statistical analysis been performed appropriately and rigorously? 

Reviewer #1: Yes

Reviewer #2: Yes

4. Have the authors made all data underlying the findings in their manuscript fully available?

Reviewer #1: Yes

Reviewer #2: Yes

5. Is the manuscript presented in an intelligible fashion and written in standard English?

Reviewer #1: Yes

Reviewer #2: Yes

6. Review Comments to the Author

**Reviewer #1**: General remarks:

The manuscript improved a lot from the previous version. Especially the Introduction and the Methods now clearly introduce the general LM concept and how it is improved in this manuscript. It is now much easier for readers to follow the here presented research. Additionally, the manuscript benefits from moving some of the figures to the Supplementary Materials and now more focus is put on the main figures. Last, the newly added limitations will help readers to assess if the unquestionable valuable approach is fitting for their research question.

While the improvements of the existing LM approach to 103 class output is undoubtfully a very valuable and interesting contribution to the field, there also might be situations and advantages of the existing 19 class output approach. One that comes to mind is an easier interpretation and communication of the results, especially for a non-scientific community. While I acknowledge that this is outside the scope of this manuscript, a very brief discussion of this could be valuable.

Specific remarks:

Line 74: I would suggest rephrasing this sentence using present tense (as in the following text in the paragraph) to highlight that these improvements are presented in this manuscript.

Line 122: While it is discussed and mentioned later in the manuscript that the classification into the three classes depends on the research questions, data availability, … it might be helpful for readers to already give a brief outlook/reference to this later in-depth discussion already here in the manuscript.

Line 222: Does the moving window have to be a square, or are also approximated circles (within the raster context) possible?

Line 266: Similar as with the previous comment, I think the latest version of the manuscript benefits from adding a discussion about the anthropogenetic impact on most natural ecosystems later in the Discussion section. Yet, I would again suggest to already briefly mention this here and reference to the Discussion.

Line 360: I cannot fully follow the reasoning about “human activity tends to be locale in scale but also spatially pervasive” as these two are rather contradicting. Furthermore, one interesting point might be that the first natural habitat could also be understood as an absence of human impacts, whereases the other two classes are a direct result of human impacts (urban and agriculture).

**Reviewer #2: **The authors conducted great work in responding to my comments, and I am satisfied with most of the edits and responses. The only thing with which I do not agree is the assumption about 'error-free reference data.' Proportion estimations and area estimations are important parts of creating and using land cover products. However, at the same time, I fully understand the authors' point regarding why they do not want to extend their experiment with additional sampling, validation, and re-calculations. It is outside the scope of their particular improvement. However, it is still very important for the authors to include in the discussion a section that describes best practices for validation, area, and proportion estimations in the context of Landscape Mosaic Models use.

7. PLOS authors have the option to publish the peer review history of their article (what does this mean?). If published, this will include your full peer review and any attached files.

Reviewer #1: No

Reviewer #2: No

---

## [Author Response · Author response to Decision Letter 1]

8 May 2024

PLoS ONE One Response to Reviewers [PONE-D-23-39102R1]

Reviewer #1: General remarks:

The manuscript improved a lot from the previous version. Especially the Introduction and the Methods now clearly introduce the general LM concept and how it is improved in this manuscript. It is now much easier for readers to follow the here presented research. Additionally, the manuscript benefits from moving some of the figures to the Supplementary Materials and now more focus is put on the main figures. Last, the newly added limitations will help readers to assess if the unquestionable valuable approach is fitting for their research question.

While the improvements of the existing LM approach to 103 class output is undoubtfully a very valuable and interesting contribution to the field, there also might be situations and advantages of the existing 19 class output approach. One that comes to mind is an easier interpretation and communication of the results, especially for a non-scientific community. While I acknowledge that this is outside the scope of this manuscript, a very brief discussion of this could be valuable.

Response: We appreciate the reviewer's clarity and specificity on actions to improve the manuscript. We added a short sentence (line 403 in the revised version) to clarify that the enhanced 103-class model also includes the information of the original 19-class model.

**Specific remarks**:

Line 74: I would suggest rephrasing this sentence using present tense (as in the following text in the paragraph) to highlight that these improvements are presented in this manuscript.

Response: Changed to present tense as suggested (line 74 in the revised version). 

Line 122: While it is discussed and mentioned later in the manuscript that the classification into the three classes depends on the research questions, data availability, … it might be helpful for readers to already give a brief outlook/reference to this later in-depth discussion already here in the manuscript.

Response: Thank you. We have now added this comment and a reference to the Discussion section (line 123 in the revised version). 

Line 222: Does the moving window have to be a square, or are also approximated circles (within the raster context) possible?

Response: Conceptionally, the moving window can be of any shape. From a practical perspective, as is often the case in spatial analysis using raster maps, the moving window (kernel) is of square shape and of size n x n pixels (in our study, n = 7, 13, 27, 81, or 243).

Line 266: Similar as with the previous comment, I think the latest version of the manuscript benefits from adding a discussion about the anthropogenetic impact on most natural ecosystems later in the Discussion section. Yet, I would again suggest to already briefly mention this here and reference to the Discussion.

Response: Thank you, we have added a reference to the Discussion section (line 266 in the revised version).

Line 360: I cannot fully follow the reasoning about “human activity tends to be locale in scale but also spatially pervasive” as these two are rather contradicting. Furthermore, one interesting point might be that the first natural habitat could also be understood as an absence of human impacts, whereases the other two classes are a direct result of human impacts (urban and agriculture).

Response: The reasoning is that human activity starts at local scale, but its influence also impacts other non-local scales. We rephrased the line accordingly in the revised version and added the observation you suggested in line 367 ff of the revised manuscript, thank you for this suggestion.

Reviewer #2

The authors conducted great work in responding to my comments, and I am satisfied with most of the edits and responses. The only thing with which I do not agree is the assumption about 'error-free reference data.' Proportion estimations and area estimations are important parts of creating and using land cover products. However, at the same time, I fully understand the authors' point regarding why they do not want to extend their experiment with additional sampling, validation, and re-calculations. It is outside the scope of their particular improvement. However, it is still very important for the authors to include in the discussion a section that describes best practices for validation, area, and proportion estimations in the context of Landscape Mosaic Models use.

Response: Thank you for this suggestion. It would have been interesting to explore this aspect more in depth. However, this would not be possible because this topic is beyond the scope of our paper. Nevertheless, as suggested, we included a new paragraph in the Discussion section addressing the suggested points (line 393 – 396 in the revised version).

---

## [Editor Report · Decision Letter 2]

9 May 2024

Revisiting the Landscape Mosaic model

PONE-D-23-39102R2

Dear Dr. Vogt,

We’re pleased to inform you that your manuscript has been judged scientifically suitable for publication and will be formally accepted for publication once it meets all outstanding technical requirements.

Kind regards,

Kristofer Lasko, PhD

Academic Editor

PLOS ONE
---

## [Editor Report · Acceptance letter]

17 May 2024

PONE-D-23-39102R2 

PLOS ONE

Dear Dr. Vogt, 

I'm pleased to inform you that your manuscript has been deemed suitable for publication in PLOS ONE. Congratulations! Your manuscript is now being handed over to our production team.

Kind regards, 

on behalf of

Dr. Kristofer Lasko 

Academic Editor

PLOS ONE